# Effect of Cashew Nutshell Extract, Saponins and Tannins Addition on Methane Emissions, Nutrient Digestibility and Feeding Behavior of Beef Steers Receiving a Backgrounding Diet

**DOI:** 10.3390/ani14213126

**Published:** 2024-10-30

**Authors:** Wilmer Cuervo, Camila Gómez, Federico Tarnonsky, Ignacio Fernandez-Marenchino, Araceli Maderal, Federico Podversich, Juan de J. Vargas, Nicolas DiLorenzo

**Affiliations:** North Florida Research and Education Center, University of Florida, Marianna, FL 32446, USA; cgomezlopez@ufl.edu (C.G.); ftarnonsky@ufl.edu (F.T.); fernandezignacio@ufl.edu (I.F.-M.); amaderal@ufl.edu (A.M.); fpodversich@ufl.edu (F.P.); juan.vargasmartinez@colostate.edu (J.d.J.V.)

**Keywords:** anacardic acid, essential oils, methane intensity, acceptability, roughage, secondary metabolites

## Abstract

Mitigation of methane emissions from livestock is crucial for addressing climate change. Beef cattle are major methane emitters, particularly during the backgrounding phase, when they are fed fibrous diets. Plant secondary metabolites (PSM) in cashew nutshell extract (CNSE), saponins, and tannins emerge as promising additives for methane mitigation. This study aimed to evaluate the effects of CNSE, saponins + tannins (ST), and their combination on methane emissions, feeding behavior, performance, intake, and digestibility in crossbred steers fed a backgrounding diet. When the additives were individually included, steer visits to the feedbunk were less frequent and for a shorter duration. After ST addition, dry matter, organic matter, and fiber intake was increased compared with CNSE. Positive responses in intake and digestibility were further enhanced with the feeding of the PSM tested (CNSEST), suggesting an interaction between the additives that improved diet acceptability and digestibility. Although CNSE did not affect methane emissions in the evaluated backgrounding diet, ST reduced enteric methane emissions for every kg of digested fiber, indicating a beneficial saponins–tannins interaction for methane mitigation without compromising animal performance. These results suggest an opportunity to explore new mixtures of PSM to be included in backgrounding diets, thereby mitigating methane emissions from livestock and promoting beef industry sustainability.

## 1. Introduction

The beef industry plays a crucial role in providing food security to the growing world population; however, beef operations are a significant contributor to greenhouse gas emissions, with enteric fermentation being the primary source of methane emissions from ruminants [1]. Methane (CH_4_) is a potent greenhouse gas, with a global warming potential 27–30 times greater than carbon dioxide. Therefore, reducing the rate of emissions and methane intensity from ruminants is a crucial step towards mitigating the impacts of climate change, considering beef cattle under grazing conditions exceed the CH_4_ intensity (CH_4_ per kg of OM fermented) reported for dairy cattle [2,3].

Cattle enteric methane emissions intensity, yield, and rate can be mitigated through several dietary strategies, among which additives containing plant secondary metabolites (PSM) such as saponins [4], tannins [5], essential oils [6], and similar plant secondary metabolites have shown effectiveness in mitigating methane [3,7]. Among the PSM, anacardic acid, cardanol, and cardol contained in the cashew nutshell extract (CNSE) have shown a reduction in the expression of the *mcrA* gene of methanogenic archaea responsible for CH_4_ production [8] as well as a significant reduction of such populations [9]. Similarly, saponins have been recognized as potent ruminal modulators, thus, extracts from tea and yucca [10] and from Saponaria [4] added to the diet significantly reduced ruminal acetate concentration and methane emitted, respectively. Likewise, there is evidence showing that hydrolyzable tannins added as extract resulted in a significant reduction in in vitro methane production [11,12,13] and the reduction of methanogens and protozoa [12,14].

Although there is evidence of enteric methane mitigation by feeding the aforementioned PSM in cattle, some authors have reported contradictory effects on enteric methane reduction when using saponins [15], tannins [16], and CNSE [17]. Moreover, the use of CNSE in cattle has been associated with a reduction in dry matter intake and low acceptability [17], while saponins and tannins added to the diet have been associated with decreased ruminal digestibility of protein and fiber [18,19,20]. Despite multiple in vitro studies being conducted, there are few in vivo studies addressing the potential effects of saponins and tannins (ST) in combination with CNSE on methane emission and nutrient digestibility. Therefore, the objective of this experiment was to investigate the potential of CNSE alone or in combination with ST on in vivo CH_4_ emissions, feeding behavior, and digestibility of beef steers fed a backgrounding diet. This research aimed to contribute to the development of sustainable and environmentally friendly livestock production systems. Our hypothesis was that the combination of CNSE and ST will exert a synergistic effect, leading to methane mitigation without affecting the digestibility of nutrients.

## 2. Materials and Methods

### 2.1. Experimental Design, Animals, and Treatments

The study utilized 16 Angus crossbred steers (347 ± 30 kg, 10 ± 1 months old) housed in a feedlot pen (2160 m^2^) at the feed efficiency of the North Florida Research and Education Center (NFREC). The experimental steers had free access to water and ad libitum intake of a backgrounding diet based on a 70:30 combination of corn silage and cottonseed burrs (Table 1). After adapting (1 month) 30 steers, sixteen steers with the greater intake and frequency of visits to the Super SmartFeed (SSF) system (C-Lock Inc., Rapid City, SD, USA) were selected. Steers were weighted and grouped (n = 4 steers/group) to obtain four groups with similar average initial body weight. Then, steers were assigned to one of four treatments in a 2 × 2 factorial arrangement: (1) no additive (NoCNSE, NoST), (2) inclusion of CNSE at 0.04% of dry matter intake (DMI; CNSE), (3) inclusion of ST (0.4% of DMI), and (4) inclusion of both CNSE and ST at the previously described rates (CNSEST; Table 2). Steers (4 per treatment) were organized in a replicated 4 × 4 Latin Square Design with four periods and four replications per treatment per period for a total of 16 experimental units per treatment.

Doses evaluated in the present studies followed the recommendation from the manufacturers of the commercial products used as sources of ST and CNSE (SDS Biotech K.K., Tokyo, Japan).

The study had four experimental periods consisting of 14 d of adaptation, 7 d of measurements, and 7 d of washout. For each experimental period, DMI, forage intake, concentrate intake, CO_2_ emission, and feeding behavior at the feedbunk were determined. Animal performance of experimental steers was assessed by measuring DMI, average daily gain (ADG), and gain to feed (G:F).

### 2.2. Diet and Additive Delivery

The basal diet (silage and cotton burrs) was delivered every morning (0800 h) in three SmartFeed automated feeders (C-Lock, Inc., Rapid City, SD, USA) at the feedlot pen to register individual DMI, frequency of visits and time at the feeder as described by Reuter et al. [22].

In order to guarantee the independence of the treatment administration and rely on every steer as the experimental unit, the additives (CNSE, ST, CNSEST) were individually delivered through the four bins of the SSF. This electronic feeding equipment automatically detected the electronic ID of each steer, delivered each treatment through an individual bin, and recorded individual feed intake and the frequency of visits. Considering the length and difficulty of the adaptation to the SSF, each steer received the additives in the same bin during the whole experiment. For each experimental period, the additives were rotated. Tag proximity sensors were configured at an approximate range of 26 cm to dispense a maximum of 2.25 kg/steer/d distributed in 11 ± 2 drops/d. Weekly calibrations were performed to confirm the administration of an accurate amount of supplement to each steer. Ground corn gluten feed was utilized as carrier, and for the non-supplemented steers, only carrier was dispensed. Despite corn gluten feed being included in Table 1, basal diet delivered through SF feedbunk only consisted of corn silage, cotton burrs, and a mineral premix. The intake of corn gluten feed (ground and as pellet) was considered in the calculation of the inclusion of ingredients (Table 2).

### 2.3. Methane Emission Determination 

Methane emission was determined using the dual hopper of the GreenFeed system (GF) (C-Lock, Inc.). The GF system allowed free movement of steers and CH_4_ was measured only when steers entered into the feeder chamber to be detected by the proximity sensor based on a radio-frequency reader. Then, between 26 and 40 L of air flowed into the collection pipe and CH_4_ was analyzed by a separate infrared analyzer using semi-pure nitrogen (zero tank) and span CH_4_ gas tank, as described by Alemu et al. [23]. Calibration of air flux sensor was performed every 30 days by releasing around 30 g of CO_2_ from a 90 g prefilled cylinder every 3 min, to obtain 100 ± 4% of gas recovery according to manufacturer’s recommendations. Similarly, the air filter was changed when air flux was below 37 L/s. Continuous data of CH_4_, CO_2_, environment temperature, pressure, airflow, and relative humidity were gathered and analyzed by the manufacturer to obtain the final CH_4_ emission data.

Pelletized corn gluten feed (up to 1.35 kg/steer/d) with an approximate diameter of 7 mm was utilized as bait through the dual hopper of the GF. Throughout the entire experimental period, daily CH_4_ and CO_2_ emissions from each steer were measured when they visited the GF, as reported by Huhtanen et al. [24]. The GF system was configured to dispense drops of bait of around 50 g at 10 s intervals up to ten times in one feeding. Steers were allowed to visit the GF system up to 10 times per day (feeding periods) with a minimum of 2 h between visits. Daily collection of raw data was processed by C-Lock Inc., and checked by head proximity, CO_2_ recovery, air and wind correction, and the duration of visits (≥2 min/visit). The main modification regarding previous studies [23,24,25] was that visits between 2 to 3 min and a minimum of 3 visits/steer/day were considered as valid data, considering the short measurement period (7 d). Invalid data (<3 visits/d) were registered for 2 animals (1 in period 3 and 1 in period 4) and were excluded from the group average calculation. Individual intake of corn gluten feed from SSF and GF was registered and included in the calculation of nutrient intake.

### 2.4. Apparent Total Tract Digestibility

Feed and feces samples were collected twice daily (0800 h and 1600 h) during the last five days of the measurement period (d 25–28) to determine the apparent total tract digestibility of dry matter (DM), organic matter (OM), crude protein (CP), neutral detergent fiber (NDF), and acid detergent fiber (ADF). Fecal samples were collected by rectal grab, while fresh samples of the carrier from each feeder bin of the SSF, corn gluten pellet from GF, and the backgrounding diet from the SmartFeed bunk were collected, weighed, recorded, dried using a forced air oven at 55 °C for 72 h and ground through a 2 mm sieve in a Wiley Mill (Arthur H. Thomas Co., Philadelphia, PA, USA). Dry fecal samples were composited by steer within period on an equal weight basis for further determination of nutrient composition and digestibility marker concentration. Indigestible NDF (iNDF) is an internal indigestible marker [26,27]. The concentration of iNDF in feces and feed was determined by incubating F57 bags (Ankom Technology Corp., Macedon, NY, USA) with 0.5 g of sample (in duplicate) inside the rumen of a cannulated steer consuming a 70:30 forage: concentrate diet for 288 h. Then, the bags were removed from the rumen, rinsed with regular tap water until runoff was clear, dried at 65 °C overnight, and incubated in the fiber analyzer following the modifications proposed by Krizsan and Huhtanen [27].

### 2.5. Feeding Behavior

The daily number of visits to the bunk, minutes spent per visit and per day, the number of meals per day and the amount of feed consumed by day, meal, and visit, as well as the duration of each visit (minutes), were obtained from the SmartFeed system (C-Lock Inc.). The amount of feed eaten per minute was registered as eating rate (g DM/min). The amount of feed Intake per visit was obtained by dividing the individual intake (DMI) by the daily number of bunk visits. The intake per min of visit was calculated by dividing DMI per visit by the average visit duration, according to the previous studies [28,29]. Visit was defined as each time the SmartFeed system detected a steer at the bunk, and a meal was considered a sum of eating periods if the gap between was not greater than 7 min [30,31].

### 2.6. Chemical Composition Determination

Basal diet (corn silage + cotton burrs) and the supplements (corn gluten feed mixed with additives) were sampled weekly to correct for D0M changes and also to adjust the formulation and the delivery on the SSF. The collected samples were dried, ground through a 2-mm screen, and composited for further chemical composition. The chemical composition of feed and feces included dry matter [32], mineral content [33], fiber, and protein. For NDF and ADF, 0.5 g of sample (feed and feces) were weighted in F57 bags (Ankom Technology Corp., Macedon, NY, USA) to perform sequential digestions with neutral and acid detergent [34] using heat-stable α-amylase according to Van Soest et al. [35], and an Ankom 200 Fiber Analyzer (Ankom Technology Corp.). For N determination to estimate crude protein, dry samples were ball-milled (25 Hz, 9 min) in a Mixer Mill MM400 (Retsch Technology, Haan, Germany) to determine the total nitrogen through the Dumas dry combustion method using a Vario Micro Cube (Elementar, Manchester, UK).

### 2.7. Calculations

Initial BW was calculated as the average unshrunk BW of steers on days −1 and 0 from each experimental period, and the final BW was the average unshrunk BW from days 20 and 21 from each period (adaptation + measurement period). Average daily gain (ADG) was determined by the difference between the final BW and the initial BW divided by 28 days.

The apparent total tract digestibility of nutrients (DM, OM, NDF, ADF, and CP) was calculated using the following formula:$$=100-\left(\;100\times \;\frac{iNDF\;concentration\;in\;feed}{iNDF\;concentration\;in\;feces}\right)\times \left(\frac{Nutrient\;concentration\;in\;feces}{Nutrient\;concentration\;in\;feed}\right)$$

Methane intensity was calculated by dividing daily emission (g) by kg of BW and kg of ADG as previously described [36,37]. Similarly, CH_4_ conversion rate was calculated according to [38] as the % of gross energy intake (GEI) following the equation reported by [39]. Additionally, the relationship between methane emission (g) and intake of NDF and energy (kg of NDF and Mcal of net energy for gain) was analyzed. Similarly, CH_4_ emission and the digestibility of main nutrients (OM, CP, NDF, ADF) were analyzed as previously reported by Ali et al. [40].

### 2.8. Statistical Analysis

Data were analyzed as a quadruplicated 4 × 4 Latin square design with a 2 × 2 factorial arrangement of treatments using the MIXED Procedure of SAS 9.4 (SAS Institute Inc., Cary, NC, USA). Carryover effects were controlled by balancing the sequence of treatments avoiding repetition of sequences. The model included CNSE and ST as main factor, their interaction, and period as fixed effects, while steer within period was included as a random effect. Statistical significance and tendencies were declared at *p* ≤ 0.05 and 0.05 < *p* ≤ 0.10, respectively. For methane emission, performance, and digestibility data, outliers were successively investigated and removed according to the Jacknife standardized residuals test [41]. For the mean separation, Tukey’s adjustment was used when a significative interaction of the main effects was detected. Results are reported as least square means and respective standard error of the mean (SEM).

## 3. Results

### 3.1. Methane Emissions

Concerning the methane emission rate (g CH_4_/d) no CNSE × ST interaction (*p* = 0.32) nor main effects of CNSE (*p* = 0.34) or ST (*p* = 0.56) were detected (Table 3). With respect to the methane yield, a CNSE × ST interaction (*p* = 0.05) revealed that individual supplementation of CNSE resulted in a greater yield (15.91 g of CH_4_/kg of DMI) when compared with CNSEST, ST, and non-supplemented steers (13.02, 13.93, and 13.22 g of CH_4_/kg of DMI, respectively).

Pertaining to methane intensity (g CH_4_/kg of BW), this variable was not affected by the main factors (*p* = 0.78 for CNSE; *p* = 0.89 for ST) nor their interaction (*p* = 0.17). A similar trend was observed for CH_4_ produced per kg of ADG. In addition to the most cited methods to determine the intensity, the amount of CH_4_ (g) per kg of apparently digested nutrients (OM, NDF, and ADF) was also calculated, with ST supplementation resulting in a reduction in this form of emissions intensity (*p* < 0.05). Thus, ST-supplemented steers produced 12.6% less CH_4_ (152.76 vs. 174.81 g) per kg of apparent digestible CP (*p* = 0.03), 23.1% less CH_4_ (84.80 vs. 110.40 g) per kg of apparent digestible ADF (*p* = 0.02) and tended to reduce 16.3% CH_4_ (37.65 vs. 45.01 g) per kg of apparent digestible NDF (*p* = 0.07), when compared to non-ST-supplemented steers.

### 3.2. Apparent Total Tract Digestibility

Regarding intake of nutrients, no interactions were observed between CNSE and ST (*p* > 0.10). No interaction on DMI from the basal diet (*p* = 0.41) nor from the corn gluten feed pellets from the GreenFeed system (*p* = 0.41) was observed (Table 4). In contrast, a CNSE×ST interaction (*p* = 0.01) revealed that steers supplemented with CNSE significantly consumed less carrier-containing additive when compared to CNSEST-supplemented steers. On average, ST-supplemented animals tended to consume an extra 1.25 kg of DM (*p* = 0.09), 0.45 kg of OM (*p* = 0.09), 0.25 kg of NDF (*p* = 0.09), and 0.1 kg of ADF (*p* = 0.1) per day when compared to steers not receiving ST. In contrast, no interaction nor main effects were detected for CP intake (1.67 kg/d; *p* > 0.1).

Regarding the apparent total tract digestibility, no interaction (*p* > 0.1) nor main effects (*p* > 0.1) were detected for DM, OM, NDF, nor ADF (58.5, 54.1, 51.5, and 47.65%, respectively). Nonetheless, a significant CNSE × ST interaction (*p* = 0.01) was detected for apparent total tract digestibility of CP, revealing that non-supplemented (60.3%) and CNSEST-supplemented steers (60.2%) did not differ in CP digestibility and were greater than ST (58.2%) and CNSE-supplemented steers (57.9%, *p* = 0.01), which might suggest that combined supplementation prevent the decrease in CP digestibility observed when individual additives were consumed.

### 3.3. Feeding Behavior

As observed in Table 5, a CNSE × ST interaction on the frequency of visits (*p* = 0.04) was detected, which revealed that non-supplemented steers (NoCNSE, NoST) had, on average 8 more visits per day when compared with ST-supplemented steers. Similarly, the CNSE × ST interaction on visit duration (min/d) revealed that steers receiving individual additives (CNSE or ST) spent on average 9 min/d less than steers receiving the combination of additives (CNSEST) or no supplementation (*p* = 0.04). No main effects (CNSE or ST supplementation) were detected on any of the feeding behavior variables evaluated.

No differences between treatments were detected in the time spent by the steers in the feed bunk (1 min), and they had around 8 meals per day, which was not different from the evaluated treatments (*p* = 0.81). A CNSE × ST interaction was observed for the total duration of the visits during the day (*p* = 0.04), which were shorter when additives were supplemented individually (33.4 min for CNSE, 34.2 min for ST) compared to the inclusion of the combination (42.3 min for CNEST) or the non-supplemented treatment (43.7 min for control). These observations coincided with a higher frequency of visits per day (*p* = 0.04) for non-supplemented (50) compared to supplemented steers (48). For the eating rate of basal diet, no interaction was detected (*p* > 0.10) nor main effects of CNSE (*p* = 0.39) nor ST (*p* = 0.76).

### 3.4. Animal Performance

No interaction (*p* = 0.29) nor main effects (*p* > 0.6) were detected for DMI as % of BW (Table 4). Similarly, no interaction (*p* = 0.37), main effects (*p* > 0.7), nor differences in ADG were observed between CNSE (1.24 kg/d), ST (1.25 kg/d), CNSEST (1.33 kg/d), and non-supplemented steers (1.29 kg/d). The gain-to-feed ratio was not affected by the inclusion of any additive (*p* > 0.10) nor their combination (*p* = 0.88).

## 4. Discussion

### 4.1. Methane Emissions

Reports suggesting the anti-methanogenic potential of anacardic acid in CNSE are not new [42], and recent evidence has corroborated this effect [8,9,17,43], which is mediated by its surfactant activity on the membrane of Archaea and Gram-positive bacteria [44,45,46] and the downregulation of methyl co-enzyme reductase subunit A (*mcrA*) gene [8]. However, in the present study, none of the additives reduced the CH_4_ emission rate (g CH_4_/d), which could partially be explained by the reduced carrier intake due to its bitter taste and lower palatability [47]. Agreeing with these observations, Branco et al. [17] attributed the lack of CH_4_ mitigation of CNSE to a lower DMI and, therefore, CNSE intake. In addition to the low carrier intake, forage intake by CNSE-supplemented steers was no different than the rest of the treatments, which could lead to a considerable fiber amount being fermented to produce methane that could not be mitigated due to insufficient CNSE. This hypothesis might be supported by Teobaldo et al. [48] who observed an increase in CH_4_ yield after the addition of low doses of CNSE (3 g/d).

In the present experiment, CNSE addition increased methane yield, whereas CNSEST reduced it to the level observed in non-supplemented steers, suggesting interactions between secondary metabolites, which seemed to exacerbate methane reduction [49]. There is compelling evidence supporting that saponins and tannins mitigate CH_4_ emission rate [48,50,51], CH_4_ yield [50,52], and CH_4_ intensity, adding higher [9,51] or lower doses [48,50] of ST than evaluated in the present study.

The reduction of CH_4_ (g) for every kg of apparently digested CP, NDF, and ADF following ST addition (ST or CNSEST) was consistent with the experiments in beef steers by Unnawong et al. [52]. Such mitigation has been attributed to the anti-protozoa activity of ST [15,53,54], reduction in DM digestibility and acetate: propionate ratio [55,56], diminished ruminal fermentation of N [15,52], and reduced NDF and CP apparent digestibility [18], particularly in high-forage diets [49]. Likewise, CNSE in CNSEST could reduce CH_4_ intensity through a shift in microbial diversity towards propionate producers such as Prevotellaceae [9]. Thus, while the chemical structures of ST could bind sterols in protozoa and Archaea membrane [57], CNSE could reduce *Butyrivibrio fibrisolvens* populations [17] known for their role in ruminal degradation of protein [58], thereby reducing CH_4_ synthesis and CP digestibility. Taking together, this evidence suggests that using ST as a feed additive in beef and dairy cattle, either alone or in combination with CNSE, promotes a shift in ruminal microbiota, leading to changes in the metabolic fate of hydrogen away from CH_4_.

### 4.2. Nutrients Intake and Apparent Total Tract Digestibility

The reduction in DMI from the carrier (Table 3) containing CNSE (Table 3) is in agreement with Gandra et al. [44], who observed a DMI reduction in dairy heifers receiving CNSL in a high-grain diet. Authors attributed this response to changes in propionate proportion and to the antimicrobial properties of essential oils. Likewise, the lack of differences in intake between supplemented and non-supplemented steers coincided with several authors that found no effect on nutrient intake using CNSL [8,48,59,60].

In contrast to the tendency for a greater DM, OM, NDF, and ADF intake from ST-supplemented steers observed here, previous studies [61,62] observed a reduction in DMI from lambs and dairy cows receiving 20 and 60 g/kg of DM in saponins and 80 g/kg, respectively. Authors suggested that elevated doses of tannins but not of saponins might have adverse effects on DMI. Unlike the present study, previous experiments reported an increase in NDF digestibility following saponins inclusion at 6–7 g/d [44,61]. Such difference might be attributed to the elevated inclusion of gin-trash, the reduced saponins concentration, or the low intake of carriers containing CNSEST in the present study.

The increased protein digestibility after CNSEST supplementation had been previously observed following saponins [63] tannins [61], their combination [49], and CNSE [44] supplementation. Some bioactive compounds present in CNSE and ST (anacardic acid, polyphenols, and tannic acids) have been reported to shift ruminal protein degradation [59], reduce protein availability for ruminal microbes [49], and generate a higher flow of protein to the duodenum [61]. This evidence might help understand why combined supplementation (CNSEST) prevented the decrease in CP digestibility observed when individual additives (CNSE or ST) were supplemented.

### 4.3. Feeding Behavior

Few works have evaluated cashew nutshell as extract (not as liquid) in growing beef cattle. The lack of changes in feed intake after ST addition coincided with previous reports [52,64], which found no effect on forage nor concentrate intake after supplementing 0.4% of saponins. In contrast, Goetz et al. [65] reported increases in the frequency of feeding after adding 5 g/d of CNSE in lactating dairy cows. These authors attributed such an increase to a higher ruminal pH derived from CNSE addition.

The lower number of visits and the shorter duration of each visit when additives were used alone (CNSE, ST) or in combination (CNSE) coincided with previous experiments using CNSE, which resulted in a reduction in feed intake [66]. On the contrary, Coutinho et al. [67] and Carvalho et al. [59] did not observe any effect on intake behavior following CNSE supplementation (7 and 2 g/d, respectively), in dairy cows or Nellore bulls. Major changes associated with CNSE included greater fiber digestibility and shifts in volatile fatty acids profile. In this sense, a greater propionic acid yield has been proposed as a trigger of a satiety mechanism [68,69].

The effects of adding CNSE alone or in combination on feeding behavior are scarce and contradictory, while in the present study, individual additives (ST, CNSE) generated shorter bunk visits duration than combined additives (CNSEST), previous studies [17,59,60,67] did not report changes in feeding behavior following the addition of CNSE. Such discrepancies might be associated with differences in the experimental design, particularly the evaluated animal model, including Zebu steers [17], dairy cows [67], beef steers [59], and finishing beef bulls [60]. Likewise, the differences when using Technical grand liquid cashew nutshell [17] or liquid CNSE [67] extract might reduce negative impacts on DMI. Similarly, the combination of CNSE with other additives (i.e., essential oils) reduced negative impacts on nutrient intake [59].

The reduced intake frequency on ST and CNSE compared to CNSEST observed here suggested a potential interaction between the active molecules on such additives. In this sense, ruminal foam derived from saponins addition was associated with a reduced particulate passage rate [70], which might affect feed intake [71]. It can be hypothesized that chemical interaction between CNSE and ST-mediated ruminal fermentation changes leading to a reduced passage contributed to feed intake cessation.

### 4.4. Animal Performance

Similar to the present study, several authors reported no changes in milk yield in dairy cattle [17,67,72], efficiency, ADG, or carcass quality in finishing beef bulls [59,73] following CNSE supplementation. Effects of CNSE on roughage diets (i.e., backgrounding) might be contradictory and rely on the combination with other chemical structures contained in the diet [74].

On the other hand, it has been reported that the formation of a tannin–protein complex leads to the reduction of digestible N [75], which potentially might lead to impaired tissue accretion [52]. Nonetheless, shifts in ruminal fermentation parameters mediated by saponins and tannins [76,77] do not correspond to the evidence of unchanged ADG, final BW, and feed efficiency [78,79,80,81], agreeing with the observed in the present experiment.

Animal performance data reported in this study corresponded to changes registered during 3 weeks (2 for adaptation and 1 for measurement) of each experimental period, which is a shorter term when compared to previous studies measuring performance parameters after CNSE [59] or ST [82] supplementation. Such variables were considered based on previous evidence from our research group [83] showing that after 1 week of CNSE and ST supplementation to cannulated beef steers, significant changes in volatile fatty acids, ammonia, and in vitro methane production were observed, suggesting potential changes in digestion and absorption process during that time span.

In the present experiment, evidence was presented of interactions between the plant secondary metabolites used as feed additives (ST and CNSE), on several response variables. While enteric CH_4_ emitted per kg of digested CP was reduced, the extent of CP digestibility was affected when ST and CNSE were individually added but was unaffected when the combination was added to the diet of beef steers. Knowing that ST has the ability to bind and inactivate lipid structures along the gastrointestinal tract of cattle [61,84] and that the chemical structure of anacardic acid in CNSE is based on phenolic lipids [85], one could suggest that secondary metabolites in ST could bind and inactivate CNSE lipid structures, affecting its ruminal effects and causing their excretion in the feces. Such interactions should be further evaluated on in vitro conditions to evaluate potential changes in ruminal fluid fermentation properties and methane production, as well as In vivo trials to evaluate their impact on digesta passage, digestibility, and intake rate on grain-based diets.

## 5. Conclusions

Steers receiving a backgrounding diet with ST at 0.4% of DMI exhibited a reduced frequency of visits to the feedbunk when compared to non-supplemented steers. Similarly, when steers received individual supplements (ST or CNSE), they spent less time per day eating at the feedbunk compared to non-supplemented steers or those fed both additives. Steers supplemented with CNSE steers consumed less carrier, leading to a lower additive intake, which coincided with a greater methane yield and lower CP digestibility. Nonetheless, the combined inclusion of additives (CNSEST) diminished CNSE’s negative effects on nutrient intake in beef cattle, suggesting a synergistic action of the saponins on Tannins and on CNSE. Neither the individual nor combined supplementation of additives leads to negative effects on the animal performance parameters evaluated in the study. Yet, those responses must be validated in further studies using longer time of exposure to the additives and their effect on animal performance.

The current study identified a reduction in methane intensity when measured in terms of g of CH_4_ per kg of digested ADF and NDF following the addition of ST. This reduction proposes a synergistic interaction between the chemical structures of ST and the fermentation and digestion of fiber. Our hypothesis is that ST added to backgrounding diets may induce shifts in ruminal fermentation, leading to the redistribution of metabolic byproducts, such as carbons and hydrogen, away from methanogenic pathways.

## Figures and Tables

**Table 1 animals-14-03126-t001:** Analyzed ^1^ chemical composition of the ingredients (DM basis) of diets fed to Angus crossbred steers receiving a backgrounding diet.

	Ingredient ^2^		
Item ^3^	CS	CGT	CGF
DM, %	47.4	90.8	96.9
CP, %	8.2	12.2	22.6
aADF, %	16.6	58.8	12.1
aNDF, %	42.9	60.5	35.0
Starch, %	32.5	1.1	16.9
Lignin, %	3.2	15.8	1.9
EE, %	3.2	3.6	3.3
TDN, %	67.6	48.5	80.0
OM, %	95.7	87.9	91.8
NEm, Mcal/kg ^4^	1.5	0.9	1.9
NEg, Mcal/kg ^4^	0.9	0.3	1.3

^1^ Dairy One Forage Testing Laboratory, Ithaca, NY. ^2^ CGT: cotton gin trash; CS: corn silage; CGF: corn gluten feed; Mix of CS and CGT as well as CGF were sampled weekly and composited throughout the study. Samples of the three additives were collected. ^3^ DM = dry matter; CP = crude protein; EE = ether extract; aADF = acid detergent fiber corrected with alpha-amylase; aNDF = neutral detergent fiber corrected with alpha-amylase; TDN = total digestible nutrients; NEm = net energy of maintenance; NEg = net energy of gain. ^4^ Estimated using the BCNRM software (NASEM 2016) [21].

**Table 2 animals-14-03126-t002:** Inclusion of ingredients and analyzed ^1^ chemical and nutrient composition (DM basis) of corn silage and cotton gin trash supplemented with corn gluten feed (ground or pelletized) fed to Angus crossbred steers.

	Treatment ^1^			
	CNSE		No CNSE	
Ingredient (% DM)	ST	No ST	ST	No ST
Corn silage ^2^	51.0	51.0	51.0	51.0
Cotton-gin trash ^2^	22.0	22.0	22.0	22.0
Corn gluten feed (ground) ^3^	16.6	17.0	16.6	17.0
Corn gluten feed (pellet) ^4^	9.96	9.96	10.0	10.0
Cashew nutshell extract ^3^	0.04	0.04	-	-
ST additive ^3^	0.4	-	0.4	-
Formulated dietary values ^5^				
DM, %	70.3	70.1	70.1	70.3
CP, % DM	12.9	13.1	12.9	13.1
NDF, % DM	44.5	44.7	44.6	44.7
ADF, % DM	19.1	18.9	18.5	19.4
NEg, Mcal/kg DM ^6^	0.9	0.9	0.9	0.9

^1^ CNSE: cashew nutshell extract; ST: commercial product with saponins and tannins. ^2^ Delivered as basal diet on the SmartFeed system. ^3^ Delivered as carrier including the additives through the Super SmartFeed system. ^4^ Delivered as bait through the GreenFeed system. ^5^ DM = dry matter; CP = crude protein; ADF = acid detergent fiber using alpha-amylase; NDF = neutral detergent fiber using alpha-amylase. ^6^ NEg = net energy of gain, estimated using the BCNRM 2016 software from NASEM (2016). Basal diet includes a commercial vitamin and mineral premix at an inclusion rate of 2% of total DM.

**Table 3 animals-14-03126-t003:** Influence of the addition of CNSE and ST on methane emission, yield, and intensity of beef steers under a backgrounding diet.

	Treatment ^1^				SEM ^3^	*p*-Value ^2^		
Variable ^4^	CNSE		No CNSE			CNSE	ST	CNSE × ST
	ST	No ST	ST	No ST				
CH_4_ emission rate, g CH_4_/d	149.14	158.53	149.43	146.94	5.92	0.34	0.56	0.32
CH_4_ yield, g/kg of DMI	12.42 ^b^	14.91 ^a^	13.14 ^b^	12.56 ^b^	0.89	0.32	0.23	0.05
Ym, %	4.08	4.68	4.13	3.75	0.32	0.17	0.72	0.13
CH_4_ intensity								
g CH_4_/kg ADG	118.16	139.76	141.86	124.08	14.29	0.78	0.89	0.17
g CH_4_/kg BW	0.351	0.381	0.352	0.341	0.01	0.14	0.33	0.14
g CH_4_/kg OMd	24.03	30.54	27.23	26.93	2.36	0.93	0.19	0.16
g CH_4_/kg CPd	152.81 ^b^	178.37 ^a^	152.71 ^b^	171.25 ^a^	9.67	0.71	0.03	0.72
g CH_4_/kg NDFd	35.79 ^X^	47.67 ^Y^	39.50 ^X^	42.32 ^Y^	3.94	0.84	0.07	0.25
g CH_4_/kg ADFd	79.77 ^b^	115.31 ^a^	89.82 ^b^	105.49 ^a^	10.43	0.99	0.02	0.34

^1^ CNSE = addition of cashew nutshell extract; ST = addition of commercial product containing saponins and tannins. Ym = methane conversion factor (% of gross energy intake; GEI). ^2^ Observed significance levels for main effects of CNSE (CNSE or No CNSE), saponins and tannins commercial product (ST or No ST) and their interaction (CNSE × ST). ^3^ Pooled standard error of treatment means (n = 16 steers/treatment). ^a,b^ Means with different superscripts within a row indicated statistical difference, *p* ≤ 0.05. ^X,Y^ Means with different superscripts tend to be different, 0.05 ≤ *p* ≤ 0.1. ^4^ DMI = Dry matter intake; GEI = Gross energy intake; OMd = organic matter digested; CPd = crude protein digested; NDFd = neutral detergent fiber digested; NDFi = NDF ingested. NEgi = Net energy for gain ingested.

**Table 4 animals-14-03126-t004:** Effect of CNSE and ST interaction on nutrient intake, apparent total tract digestibility, and animal performance in beef steers receiving a backgrounding diet.

	Treatment ^1^					*p*-Value ^2^		
	CNSE		No CNSE		SEM ^3^	CNSE	ST	CNSE × ST
Item ^4^	ST	No ST	ST	No ST				
Intake, kg/d DM basis								
DM	12.78 ^X^	11.91 ^Y^	11.79 ^X^	12.34 ^Y^	0.74	0.33	0.09	0.89
Basal diet *	8.91	8.41	8.12	8.71	0.77	0.75	0.96	0.41
Carrier **	2.21 ^a^	1.91 ^b^	2.11 ^ab^	2.14 ^ab^	0.07	0.83	0.13	0.01
Bait ***	1.61	1.62	1.54	1.42	0.11	0.68	0.90	0.08
OM	11.90	11.31	11.17	11.62	0.50	0.34	0.09	0.89
CP	1.72 ^X^	1.75 ^X^	1.62 ^Y^	1.73 ^X^	0.05	0.48	0.12	0.7
NDF	5.89 ^X^	5.52 ^Y^	5.48 ^X^	5.70 ^Y^	0.27	0.36	0.09	0.88
ADF	2.39	2.22	2.21	2.38	0.12	0.31	0.11	0.63
Digestibility, %								
DM	58.51	58.42	58.50	58.79	1.31	0.85	0.95	0.89
OM	54.02	54.32	54.31	53.65	1.35	0.93	0.90	0.75
CP	60.21 ^a^	57.90 ^b^	58.20 ^b^	60.29 ^a^	0.24	0.76	0.94	0.01
NDF	50.69	51.79	51.78	51.59	0.69	0.51	0.53	0.31
ADF	47.20	48.22	47.21	48.02	0.79	0.87	0.26	0.86
Animal Performance								
DMI, % BW	2.93	2.74	2.72	2.82	1.41	0.63	0.74	0.29
ADG, kg/d	1.33	1.24	1.25	1.29	0.08	0.87	0.76	0.37
G:F, kg/kg	0.11	0.11	0.11	0.12	0.01	0.7	0.76	0.88

^1^ CNSE = addition of cashew nutshell extract; ST = addition of commercial product containing saponins and tannins. ^2^ Observed significance levels for main effects of CNSE (CNSE or No CNSE), saponins and tannins commercial product (ST or No ST) and their interaction (CNSE × ST). ^3^ Pooled standard error of treatment means, n = 16 steers/treatment. ^a,b^ Means with different superscripts within a row indicated statistical difference, *p* ≤ 0.05. ^X,Y^ Means with different superscripts tend to be different, 0.05 ≤ *p* ≤ 0.1. ^4^ DM = dry matter; OM = organic matter; CP = crude protein; NDF = neutral detergent fiber; ADF = acid detergent fiber. BW = body weight; ADG = average daily gain; DMI = dry matter intake; G:F = gain-to-feed ratio. * Delivered on the SmartFeed system. ** Delivered through the Super SmartFeed system. *** Corn gluten feed pellet delivered as bait through the GreenFeed system.

**Table 5 animals-14-03126-t005:** Influence of CNSE and ST addition on feeding behavior in beef steers consuming a backgrounding diet *.

	CNSE		No CNSE			*p*-Value ^1^		
Item	ST	No ST	ST	No ST	SEM ^2^	CNSE	ST	CNSE × ST
Bunk visits frequency, visits/d	48.79 ^ab^	43.90 ^ab^	41.74 ^b^	50.03 ^a^	2.22	0.88	0.60	0.04
Bunk visits length, min/BV	1.14	1.02	1.04	0.95	0.12	0.75	0.64	0.81
Bunk visits duration, min/d	42.32 ^a^	33.41 ^b^	34.24 ^b^	43.71 ^a^	4.34	0.80	0.94	0.04
Daily meals, meals/d	8.32	7.84	7.92	8.51	0.43	0.79	0.94	0.34
Meal size, g DM/meal	1078.80	1074.89	940.09	1048.12	88.21	0.51	0.68	0.66
Intake per visit, g/visit	157.52	181.40	170.11	164.79	7.90	0.85	0.41	0.21
Eating rate, g DM/min	285.70	304.69	270.89	269.39	20.2	0.39	0.76	0.72
Dry matter intake, kg/d *	7.94	8.02	7.31	8.30	0.71	0.79	0.38	0.51

^1^ Observed significance levels for main effects of CNSE (CNSE or No CNSE), saponins, and tannins commercial product (ST or No ST) and their interaction (CNSE × ST). ^2^ Pooled standard error of treatment means, n = 16 steers/treatment. ^a,b^ Means with different superscripts within a row indicated statistical difference, *p* ≤ 0.05. CNSE = addition of cashew nutshell extract; ST = addition of commercial product containing saponins and tannins; BV = Bunk Visits. * Represent DMI from basal diet (corn silage + gin trash) delivered through SmartFeed.

## Data Availability

The data presented in this paper are available on request from the corresponding author.

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
