# Peer review of "Effect of Cashew Nutshell Extract, Saponins and Tannins Addition on Methane Emissions, Nutrient Digestibility and Feeding Behavior of Beef Steers Receiving a Backgrounding Diet"

_animals, 2024, doi:10.3390/ani14213126_

Round 1
Reviewer 1 Report
Comments and Suggestions for Authors
The article is well written and represents a scientific contribution.
However, the authors carried out a very short study period, which may have interfered with the results.
Why didn't they extend the trial period?
In the discussion part, the authors began by reporting eating behavior.
We suggest that you start according to the sequence of results.
The conclusions are pertinent, since the authors recognize that the experiment was carried out in a short period of time and that new studies on the topic are necessary. Mainly related to the time taken to carry out the study.
Author Response
Comment 1: The article is well written and represents a scientific contribution. However, the authors carried out a very short study period, which may have interfered with the results.
Response 1: Thank you for pointing out this out. Indeed, each experimental period was relatively short (28 days) compared to previous studies evaluating methane (between 42 to 70 d). By using a Latin square design and a factorial arrangement we wanted to increase the statistical power of our experiment. The length of the measurement period is based on previous In vitro and Ex vivo experiments performed by our research group, in which after 7 and 14 days of adding CNSE, we observed significant changes in VFA and in vitro CH4 production per gr of fermented organic matter.
Comment 2: Why didn't they extend the trial period?
Response 2: this is an important question to the methodology of the study. Several factors influenced the duration of the study. In addition to the biological factor stated in the previous response, we utilized a group of commercial steers that should finish their normal productive cycle in the station before heading to the slaughterhouse, therefore, the study had to be performed before this time. Similarly, the feedlot facilities, feeder, and Super SmartFeed utilized included in the experimental arrangement have a continuous schedule of usage, and the availability to use the experimental animals, the evaluated diets and the equipment for this experiment during my PhD program was limited to those 4.5 months.
Comment 3: In the discussion part, the authors began by reporting eating behavior. We suggest that you start according to the sequence of results.
Response 3: We agree with this comment. Therefore, we organized the results and the discussion in the same order in which we presented them in the Materials and Methods section.
Comment 4: The conclusions are pertinent, since the authors recognize that the experiment was carried out in a short period of time and that new studies on the topic are necessary. Mainly related to the time taken to carry out the study.
Response 4: The reviewer raises a major limitation of the study, and we completely agree with this comment. Previous in vitro and Ex vivo studies performed by our research group (unpublished studies) showed that even after 7 days of adding CNSE, or ST, ruminal VFA profile, in vitro CH4, and protozoa population were changed, suggesting shifts in ruminal fermentation.
We share the reviewer’s concern about the available time to evaluate performance parameters. With animal performance variables we intended to obtain information about potential effects of CNSE and ST on productive traits. However, we are aware that the influence of the prolonged exposure to CNSE on animal performance must be further addressed under extended periods of time (ideally 56 to 72 days) to obtain comparable values previously reported. In addition, the idea behind considering some performance variables was to analyze the potential deleterious effects of the additives when supplementing under in vivo conditions to growing beef cattle

Reviewer 2 Report
Comments and Suggestions for Authors
General comments
Certainly, we should avoid using the term 'palatability' when discussing ruminants such as cattle and instead refer to 'acceptability' to describe their preference or willingness to consume certain feeds.
The authors mention that after adapting 30 steers for one month, 16 steers with the greatest intake and frequency of visits to the Super SmartFeed (SSF) system were selected and 'grouped by bodyweight (BW).' However, it is unclear what is meant by 'grouped by bodyweight' in this context. Further clarification is needed regarding the criteria or method used for grouping the animals based on their bodyweight, as this could influence the interpretation of the treatment effects.
Considering that the data were analyzed using a quadruplicated 4 × 4 Latin square design with a 2 × 2 factorial arrangement of treatments, I am concerned about the potential long-term responses in the animals' body condition, particularly in terms of DMI, average daily gain, and gain-to-feed ratio. While short-term effects are evident, it remains unclear how sustained exposure to the treatments may influence these performance indicators over a prolonged period.
The authors state that 'The basal diet (silage and cotton burrs) was delivered every morning in Smart Feed automated feeders, with ground corn gluten feed used as a carrier, and for the non-supplemented steers, only the carrier was dispensed. However, it is important to note that the diet composition also included corn gluten feed in pellet form (Table 2) . This creates a potential inconsistency in how the basal diet and the carrier are described, and clarification is needed to avoid confusion regarding the form and role of corn gluten feed in the diet.
It is important to note that Table 2 does not report the inclusion of any mineral or vitamin mixtures in the diet composition. Given the significance of these supplements in ensuring a balanced diet for optimal animal performance, it would be helpful to clarify whether these components were included but not reported, or if they were excluded from the formulation.
The authors have chosen to present the Methods and Materials (M&M), Results, and Discussion as distinct sections. However, it would be beneficial for the order of these topics to remain consistent across all sections.
Is not clear the ST composition.
Specific comments
L31-32 Sixteen steers in a 4 x 4 LSD, please indicated the replications of LSD
Keywords: palatability; Consider my previus comment about this term
L50: 30 or 28 times? please confirm and support this data
L55: please specify wich specie or system
L64-65: There are two "types" of tannins, is not clear if this sentence is about both. In addition, in abstract is not clear which type of tannin was used in this study.
L68-70: In cattle?
L71-72: Be careful, Osimari et al., 2017 reported that CNSE had no effects on intake
L90: As reported in abstract, please addad the ratio of corn silage and cottonseed burrs
L92-93: Whan meaning this grouping? please explain how were assigned steers, considering BW?
Table 2 please add the DM %
Line 232: How was BW grouping reported previusly, and considered in the mathematical model during statistical analysis?
L234: How carry-over effects were treated?
L235: Why daily methane data were not considered as repeated measured over time?
L177-189: Is not clear how the total fecal production per steer was measured or stimated.
L194-198: Was the % of dry matter in the diet considered to correct these values?
Please expand the decimal places of the p-values reported in Tables 3 to 6 to three decimal places
Authors are encouraged to standardize the number of decimal places reported in the mean and SEM values ​​in the tables 3 to 6.
L 271-274: as mean values of ST and No ST are not reported in the table, these values should be included here.
L276-278: Total dry matter intake should also be expressed as a % of the steers body weight.
The authors state that statistical significance and tendencies were declared at p ≤ 0.05 and 0.05 < p ≤ 0.10, respectively. However, in some tables, the tendencies of interaction are not accompanied by the appropriate letter superscripts to indicate these tendencies.
Considering the information content presented, Tables 4 and 6 could be merged to create a more streamlined presentation of the data.
L366-367: ST is not only saponins
L367-369: how might the higher ruminal pH resulting from CNSE addition influence feeding behavior in this context?
L375: "greater fiber digestibility" This happened in your study?
L378-381: It would be beneficial for the authors to discuss possible reasons for these discrepancies, such as differences in experimental design, dosages, or animal characteristics.
L423-430: The authors should consider discussing potential carry-over effects associated with the use of CNSE and ST in the experimental design. Given that the study employed a short-term measurement period, there is a possibility that previous treatments could influence subsequent responses, particularly in parameters such as DMI and animal performance. Carry-over effects may occur if the physiological or metabolic adaptations induced by the initial supplementation persist beyond the experimental period. This could lead to either an underestimation or overestimation of the true effects of the additives on animal performance and feeding behavior.
L436: the reduction in DMI from the carrier containing CNSE raises important considerations regarding the effective dosage of CNSE administered to the steers. Since the overall intake of the carrier was lower, the actual amount of CNSE ingested by the animals may have been insufficient to exert the expected anti-methanogenic effects. This point need to be discussed.
Author Response
Comment 1: Certainly, we should avoid using the term 'palatability' when discussing ruminants such as cattle and instead refer to 'acceptability' to describe their preference or willingness to consume certain feeds.
Response 1: Thank you for pointing out this out. We agree with this comment. Therefore, we have made the change by correcting the word palatability across the whole document.
Comment 2: The authors mention that after adapting 30 steers for one month, 16 steers with the greatest intake and frequency of visits to the Super SmartFeed (SSF) system were selected and 'grouped by bodyweight (BW).' However, it is unclear what is meant by 'grouped by bodyweight' in this context. Further clarification is needed regarding the criteria or method used for grouping the animals based on their bodyweight, as this could influence the interpretation of the treatment effects.
Response 2: Thank you for pointing out this out. We would like to clarify that the initial selection (16 out of 30) was done based on higher frequency of visits to the SSF. Once we selected 16 steers, we weight them and sort the animals to obtain 4 groups with similar average bodyweight per group (305.9, 298.2, 307.9, 310.2). The clarification is now included in the text.
Comment 3: Considering that the data were analyzed using a quadruplicated 4 × 4 Latin square design with a 2 × 2 factorial arrangement of treatments, I am concerned about the potential long-term responses in the animals' body condition, particularly in terms of DMI, average daily gain, and gain-to-feed ratio. While short-term effects are evident, it remains unclear how sustained exposure to the treatments may influence these performance indicators over a prolonged period.
Response 3: We agree with this comment and the reviewer’s reflection in this matter. The study was initially designed as a pilot project to evaluate the in vivo supplementation of CNSE, given our research group have already tested the in vitro and Ex vivo effects of CNSE. Considering in those previous studies, tannins and saponins have also promoted the reduction of in vitro CH4 production (unpublished data), we aimed to evaluate CNSE, saponins + tannins, the combination (CNSE + tannins + saponins) and a control (4 treatments). In addition, before the initial date of the experiment, only 30 steers with similar age and BW were available in the station. Moreover, experimental animals could only be used for 5 to 6 months, because they had to complete their productive cycle to be sold. So, counting 30 days of adaptation, only 4 to 5 months were available to complete the study. For all those reasons, the best experimental design available for the project was a 4 x 4 Latin square, using 4 weeks for each period.
We share the reviewer’s concern about the available time to evaluate performance parameters. With the inclusion of the analysis of such parameters we intended to obtain additional information about the effects of CNSE addition on productive traits. However, we are aware that the influence of the prolonged exposure to CNSE on animal performance must be further addressed under extended periods of time (ideally 56 to 72 days) to obtain comparable values previously reported.
Comment 4: The authors state that 'The basal diet (silage and cotton burrs) was delivered every morning in Smart Feed automated feeders, with ground corn gluten feed used as a carrier, and for the non-supplemented steers, only the carrier was dispensed. However, it is important to note that the diet composition also included corn gluten feed in pellet form (Table 2) . This creates a potential inconsistency in how the basal diet and the carrier are described, and clarification is needed to avoid confusion regarding the form and role of corn gluten feed in the diet.
Response 4: We agree with this comment since the information might be presented in a manner that is hard to follow. In this regard, on Table 2 in the superscripts 2 and 3 was indicated that ground corn gluten feed was delivered as carrier through the SSF while and corn gluten feed pellet was delivered as bait through the GreenFeed system. Corn gluten feed composition was included in table 1 as a reference to the product used as vehicle (SSF) and as bait (GF). The proportions of inclusion (table 2) for corn gluten feed corresponded to the daily amount delivered by SSF and GF. Following the suggestion from the reviewer we included a text to clarify this situation and is located at the end of the numeral 2.2. Diet and additive delivery.
Comment 5: It is important to note that Table 2 does not report the inclusion of any mineral or vitamin mixtures in the diet composition. Given the significance of these supplements in ensuring a balanced diet for optimal animal performance, it would be helpful to clarify whether these components were included but not reported, or if they were excluded from the formulation.
Response 5: Authors thank to the reviewer for raising up this point. A commercial Min&Vit premix was included and mixed with the basal diet (corn silage + cotton burrs) at 2% of DM mix. This commercial premix is commonly included when a backgrounding diet is mixed for experimental and commercial animals housed in the facilities of the research station. The value was unintentionally not reported. Following the valuable suggestion, we are adding an explanatory note at the bottom of the table 1.
Comment 6: The authors have chosen to present the Methods and Materials (M&M), Results, and Discussion as distinct sections. However, it would be beneficial for the order of these topics to remain consistent across all sections.
Response 6: We agree with this comment. Therefore, we organized the results and the discussion in the same order in which we presented in the material and methods section.
Comment 7: Is not clear the ST composition.
Response 7: To evaluate the mix of saponins and tannins (ST) we utilized a commercial product. Unfortunately, the specific composition was not disclosed by the company.
Specific comments
Comment 8: L31-32 Sixteen steers in a 4 x 4 LSD, please indicated the replications of LSD
Response 8:. We agree with this comment. Therefore, in the mentioned line we have included the number of replications (4 steers, 4 treatments, 4 periods). Basically. it is a quadruplicated Latin square design.
Comment 9:
Keywords: palatability; Consider my previous comment about this term
Response 9: We agree with this comment. As we mentioned before, the change was made across the whole manuscript.
Comment 10:
L50: 30 or 28 times? please confirm and support this data
Response 10 : Thank you for pointing out this out. The information came from the EPA and the correct number is 27 – 30 times, so it was corrected accordingly in the same line.
Environmental Protection Agency. (2021). Understanding global warming potentials. Retrieved from https://www.epa.gov/ghgemissions/understanding-global-warming-potentials
Comment 11: L55: please specify which specie or system
Response 11 : We agree with this comment. The line was changed to “Enteric Methane emission from ruminants..”
Comment 12: L64-65: There are two "types" of tannins, is not clear if this sentence is about both. In addition, in abstract is not clear which type of tannin was used in this study.
Response 12 : Agree with the comment. Therefore, we redact the line to clarify it. As mentioned in comment 7 about the tannins type, unfortunately, the specific composition was not disclosed by the company
Comment 13: L68-70: In cattle?
Response 13 : Agree, we add “Is evidence of enteric methane mitigation in cattle”
Comment 14: L71-72: Be careful, Osimari et al., 2017 reported that CNSE had no effects on intake
Response 14 : Completely agree, Osimari et al., reported changes in ruminal digestibility of protein but not intake of nutrients. The reference was removed from the sentence
Comment 15: L90: As reported in abstract, please addad the ratio of corn silage and cottonseed burrs
Response 15 : Agree, we changed to “...diet based on a 70:30 combination of corn silage and cottonseed burrs”
Comment 16: L92-93: Whan meaning this grouping? please explain how were assigned steers, considering BW?
Response 16 : The reviewer raises an important point. We would like to clarify that the initial selection (16 out of 30) was done based on higher frequency of visits to the super smart feed system. Once we selected 16 steers, we weight them and sort the animals to obtain 4 groups with similar initial average bodyweight per group (305.9, 298.2, 307.9, 310.2). The clarification is now included in the text.
Comment 17: Table 2 please add the DM %
Response 17 : Agree, The value (70.3) was added in new row of table 2.
Comment 18: Line 232: How was BW grouping reported previusly, and considered in the mathematical model during statistical analysis?
Response 18 : After the initial selection of the steers, the sorting by BW was done to obtain homogeneous groups. BW at d0 was not included as a covariate in the model since it could only be included in the analysis of the first experimental period. Considering that average initial weights (d0 of the 1st period) for the four treatments were homogeneous (305.9 kg, 298.2 kg, 307.9 kg, and 310.2 kg) we decided not to include this variable as covariate in the model. Moreover, We were concerned about include BW at d0 for a potential multicollinearity problem, since BW in subsequent periods (d30, d60, d90) were correlated with the initial BW. We consider that BW values over time are not independent; animals that start heavier tending to stay heavier, making the inclusion of the same variable (initial BW) across periods potentially problematic. In addition, some of the values (DMI as % of BW, CH4 per kg of ADG) were already including BW, confirming the potential multicollinearity if BW d0 would be included in the model. Finally, when including initial BW as a covariate in the model, it was not significant for Methane emissions (g/d; P = 0.86), concentrate intake (P = 0.90), nor total DMI (kg/d; P = 0.10) for period 1.
Comment 19: L234: How carry-over effects were treated?
Response 19 : Carryover effects were minimized by balancing the sequence of treatments avoiding repetition of sequences, and by using a period of 7 days of washout followed by a 14-d adaptation period to treatment before measurements.
Comment 20: L235: Why daily methane data were not considered as repeated measured over time?
Response 20 : The reviewer raises an interesting point here. We consider that the LSD inherently accounted for treatment, period, and steer effects, by averaging daily CH4 emissions within each period, we obtained a single value of CH4/steer/period. The treatment effect in our case was evaluated across periods and not within period as repeated measures. We also consider that modeling daily methane data as repeated measures would result in overfitting and introduce unnecessary complexity, particularly given the relatively short duration of the evaluation period (7 days). Perhaps even more importantly, the use of technology such as GreenFeed requires several visits over time in order to build confidence in the value of emissions reported. Some recent studies have addressed this issue (Dressler et al., 2023; JAS doi.org/10.1093/jas/skad176) and raise the question of what the minimum number of visits that is required for an accurate value of emissions.
Comment 21: L177-189: Is not clear how the total fecal production per steer was measured or stimated.
Response 21 : In the lines we did not mention total fecal collection. Indeed, to clarify the procedure we included in the 177 line “Fecal samples were collected by rectal grab”.
Comment 22: L194-198: Was the % of dry matter in the diet considered to correct these values?
Response 22 : This a valuable point to clarify. Besides the feed samples collected for the apparent total tract digestibility procedure, we collected weekly samples of basal diet, ground corn gluten feed (From SSF), and corn gluten feed pellets (from GF), and DM was determined to correct the DM content of the diet for each experimental period.
Comment 23: Please expand the decimal places of the p-values reported in Tables 3 to 6 to three decimal places
Response 23 : We increased the decimal numbers on the referenced tables.
Comment 24: Authors are encouraged to standardize the number of decimal places reported in the mean and SEM values ​​in the tables 3 to 6.
Response 24 : We agree, we corrected the issue, and the SEM values have 2 decimals in all tables.
Comment 25: L 271-274: as mean values of ST and No ST are not reported in the table, these values should be included here.
Response 25 : We agree, since those values were not in the table, they were included in parenthesis within the text.
Comment 26: L276-278: Total dry matter intake should also be expressed as a % of the steers body weight.
Response 26 : We could not find this reference in lines 276 – 278, however, DMI as % of BW is included in table 6.
Comment 27: The authors state that statistical significance and tendencies were declared at p ≤ 0.05 and 0.05 < p ≤ 0.10, respectively. However, in some tables, the tendencies of interaction are not accompanied by the appropriate letter superscripts to indicate these tendencies.
Response 27 : thank you for pointing this out. We agree with this comment. Therefore, we have made the change and include X, Y for the tendencies in the tables.
Comment 28: Considering the information content presented, Tables 4 and 6 could be merged to create a more streamlined presentation of the data.
Response 28 : Thank you for pointing out this out. We agree with this comment. Therefore, we have made the change and merged table 4 and 6
Comment 29: L366-367: ST is not only saponins
Response 29 : This is an important point. We agree with this comment. Therefore, we have included statement in this line to clarify.
Comment 30: L367-369: how might the higher ruminal pH resulting from CNSE addition influence feeding behavior in this context?
Response 30 : The line 367 contained “These authors attributed such increase to a higher ruminal pH derived from CNSE addition” and refers to the explanation proposed by the cited authors (Goetz et al., 2023) in which a higher ruminal pH was associated with improved rumen health, a more effective feed digestion and therefore a greater DMI.
Comment 31: L375: "greater fiber digestibility" This happened in your study?
Response 31 : The reviewer makes a great point here. Indeed, besides the interaction I CP digestibility, we did not observed changes in nutrients digestibility. Therefore, we removed the “greater fiber digestibility” from the phrase, considering the message we intended to deliver was that among the changes in ruminal fermentation promoted by CNSE, a shift in VFA profile could be associated with effects on intake behavior.
Comment 32: L378-381: It would be beneficial for the authors to discuss possible reasons for these discrepancies, such as differences in experimental design, dosages, or animal characteristics.
Response 32 : Thank you for pointing out this out. We agree with this comment. Therefore, we include in the aforementioned lines a discussion suggesting the reasons for the discrepancies between our results and previous studies.
Comment 33: L423-430: The authors should consider discussing potential carry-over effects associated with the use of CNSE and ST in the experimental design. Given that the study employed a short-term measurement period, there is a possibility that previous treatments could influence subsequent responses, particularly in parameters such as DMI and animal performance. Carry-over effects may occur if the physiological or metabolic adaptations induced by the initial supplementation persist beyond the experimental period. This could lead to either an underestimation or overestimation of the true effects of the additives on animal performance and feeding behavior.
Response 33 : Previous in vitro and Ex vivo studies performed by our research group (unpublished studies) showed that even after 3 days of removing the additive (CNSE, or ST) ruminal VFA profile, in vitro CH4, and protozoa population and viability were similar to the non-supplemented ruminal fluid, suggesting a return to normal conditions after the removal of the additives. Therefore, 7 days of washout was deemed sufficient time to assure ruminal fermentation, and digestion would return to previous conditions. Moreover, besides the washout period, each experimental period includes 2 weeks of adaptation, after which the effect of the new treatment would be independent of the previous treatment. We share the reviewer’s concern about the available time to evaluate performance parameters. With animal performance variables we intended to obtain information about potential effects of CNSE and ST on productive traits. However, we are aware that the influence of the prolonged exposure to CNSE on animal performance must be further addressed under extended periods of time (ideally 56 to 72 days) to obtain comparable values previously reported.
Comment 34: L436: the reduction in DMI from the carrier containing CNSE raises important considerations regarding the effective dosage of CNSE administered to the steers. Since the overall intake of the carrier was lower, the actual amount of CNSE ingested by the animals may have been insufficient to exert the expected anti-methanogenic effects. This point needs to be discussed.
Response 34 : The reviewer pointed out a very interesting comment. Indeed, there were some days in which animals receiving CNSE exhibited a lower intake of the vehicle and most likely they did not hit the target (0.04% of DMI). We did not expect this result, especially considering we utilized a very palatable vehicle (corn gluten feed). The experiment involved the configuration of SSF to deliver 2.25 kg/steer/d, while the observed intake of carrier was 2.21, 1.92, 2.12, and 2.14 kg/steer/d for CNSE-ST, CNSE, ST, NoCNSE-NoST, respectively. These amounts corresponded to 98%, 84%, 93%, and 93% of the expected target, meaning that the animals receiving CNSE-ST addition, consumed in average 49.6 of ST instead 50.8 g/d, and consumed 4.97 instead 5.08 g/d of CNSE. While the steers receiving the addition of CNSE alone, consumed 4.1 g/steer/d instead of the expected 4.7 g/steers/d. In this sense, previous in vitro and Ex vivo studies performed by our research group (unpublished studies) showed that the addition of 4 g of CNSE promoted changes in propionic concentration, Acetate:propionate ratio, and in vitro CH4 production per g of organic matter fermented.

Reviewer 3 Report
Comments and Suggestions for Authors
The study evaluated the potential of cardol alone or combined with saponins and tannins on methane emissions, digestibility and productive performance of beef steers.
The study has clear objectives and an adequate methodology.
However, it has some limitations:
In this regard, none of the additives (CNSE or ST) effectively reduced methane emissions. This is contrary to previous reports suggesting anti-methanogenic properties of these additives. This is addressed by the authors and attributed to the lower consumption of the additives caused by their lower palatability. Since this is the core of the study, it would be convenient to expand the explanation on this aspect.
On the other hand, the number of experimental units in each treatment was low. Nevertheless, the statistical analysis is correct. As a suggestion for future research, if there is a small number of animals, perhaps fewer treatments would be appropriate. For example, only evaluate the effect of the STs, to ensure a larger number of experimental units.
Finally, the duration of the study was short, and may not have been sufficient to capture the effects of additives on productive behavior, nutrient digestibility and methane emissions. The authors address this point in their conclusions and suggest that more research is needed on this topic with longer durations. Although this was a limitation of the study, it provides a guideline for further research on this topic.
Author Response
Comment 1: In this regard, none of the additives (CNSE or ST) effectively reduced methane emissions. This is contrary to previous reports suggesting anti-methanogenic properties of these additives. This is addressed by the authors and attributed to the lower consumption of the additives caused by their lower palatability. Since this is the core of the study, it would be convenient to expand the explanation on this aspect.
Response 1 : The reviewer raises an essential matter with this. We agree with this comment. Indeed, there were some days in which animals receiving CNSE had a lower intake of the vehicle and most likely they did not hit the target (0.04% of DMI). We did not expect this, especially considering we utilized a very palatable vehicle (corn gluten feed). The experiment involved the configuration of SSF to deliver 2.25 kg/steer/d, while the observed intake of carrier was 2.21, 1.92, 2.12, and 2.14 kg/steer/d for CNSE-ST, CNSE, ST, NoCNSE-NoST, respectively. These amounts corresponded to 98%, 84%, 93%, and 93% of the expected target. , meaning that the animals receiving CNSE-ST addition, consumed in average 49.6 of ST instead 50.8 g/d, and consumed 4.97 instead 5.08 g/d of CNSE. While the steers receiving the addition of CNSE alone, consumed 4.1 g/steer/d instead the expected 4.7 g/steers/d. In this sense, previous in vitro and Ex vivo studies performed by our research group (unpublished studies) showed that the addition of 4 g of CNSE promoted changes in propionic concentration, and in vitro CH4 production per g of organic matter fermented. Despite the astringent property of CNSE and the bitter flavor associated with tannins and saponins, it is possible that when both additives were combined they either masked each other’s unpalatable properties or balanced the sensory perception in a way that made the combined supplement more acceptable to the animals. Furthermore, it is possible that both additives when combined, may have experienced a different or more neutral physiological and sensory response, encouraging them to maintain more regular feeding patterns. We understand this is purely speculation, but we do not have many plausible explanations.
Comment 2: On the other hand, the number of experimental units in each treatment was low. Nevertheless, the statistical analysis is correct. As a suggestion for future research, if there is a small number of animals, perhaps fewer treatments would be appropriate. For example, only evaluate the effect of the STs, to ensure a larger number of experimental units.
Response 2: We agree with this comment and suggestion, considering the few steers adapted to the SSF system, it would be useful to re-evaluate the inclusion of more than 2 treatments with this design and think in a crossover with 2 treatments and longer measurement periods.
Comment 3: Finally, the duration of the study was short, and may not have been sufficient to capture the effects of additives on productive behavior, nutrient digestibility and methane emissions. The authors address this point in their conclusions and suggest that more research is needed on this topic with longer durations. Although this was a limitation of the study, it provides a guideline for further research on this topic.
Response 3: The reviewer raises one of the limitations of the study, and we completely agree with this comment. Previous in vitro and Ex vivo studies performed by our research group (unpublished studies) showed that even after 7 days of adding CNSE, or ST, ruminal VFA profile, in vitro CH4, and protozoa population were changed, suggesting shifts in ruminal fermentation.
We share the reviewer’s concern about the available time to evaluate performance parameters. With animal performance variables we intended to obtain information about potential effects of CNSE and ST on productive traits. However, we are aware that the influence of the prolonged exposure to CNSE on animal performance must be further addressed under extended periods of time (ideally 56 to 72 days) to obtain comparable values previously reported. In addition, the idea behind considering some performance variables was to analyze the potential deleterious effects of the additives when supplementing under in vivo conditions to growing beef cattle.
